# Near-optimal Anomaly Detection in Graphs using Lovász Extended Scan Statistic

**James Sharpnack**
Machine Learning Department
Carnegie Mellon University
Pittsburgh, PA 15213
jsharpna@gmail.com

**Akshay Krishnamurthy**
Computer Science Department
Carnegie Mellon University
Pittsburgh, PA 15213
akshaykr@cs.cmu.edu

**Aarti Singh**
Machine Learning Department
Carnegie Mellon University
Pittsburgh, PA 15213
aarti@cs.cmu.edu

## Abstract

The detection of anomalous activity in graphs is a statistical problem that arises in many applications, such as network surveillance, disease outbreak detection, and activity monitoring in social networks. Beyond its wide applicability, graph structured anomaly detection serves as a case study in the difficulty of balancing computational complexity with statistical power. In this work, we develop from first principles the generalized likelihood ratio test for determining if there is a well connected region of activation over the vertices in the graph in Gaussian noise. Because this test is computationally infeasible, we provide a relaxation, called the Lovász extended scan statistic (LESS) that uses submodularity to approximate the intractable generalized likelihood ratio. We demonstrate a connection between LESS and maximum a-posteriori inference in Markov random fields, which provides us with a poly-time algorithm for LESS. Using electrical network theory, we are able to control type 1 error for LESS and prove conditions under which LESS is risk consistent. Finally, we consider specific graph models, the torus, $k$-nearest neighbor graphs, and $\epsilon$-random graphs. We show that on these graphs our results provide near-optimal performance by matching our results to known lower bounds.

## 1 Introduction

Detecting anomalous activity refers to determining if we are observing merely noise (business as usual) or if there is some signal in the noise (anomalous activity). Classically, anomaly detection focused on identifying rare behaviors and aberrant bursts in activity over a single data source or channel. With the advent of large surveillance projects, social networks, and mobile computing, data sources often are high-dimensional and have a network structure. With this in mind, statistics needs to comprehensively address the detection of anomalous activity in graphs. In this paper, we will study the detection of elevated activity in a graph with Gaussian noise.

In reality, very little is known about the detection of activity in graphs, despite a variety of real-world applications such as activity detection in social networks, network surveillance, disease outbreak detection, biomedical imaging, sensor network detection, gene network analysis, environmental monitoring and malware detection. Sensor networks might be deployed for detecting nuclear substances, water contaminants, or activity in video surveillance. By exploiting the sensor network structure

(based on proximity), one can detect activity in networks when the activity is very faint. Recent theoretical contributions in the statistical literature[1, 2] have detailed the inherent difficulty of such a testing problem but have positive results only under restrictive conditions on the graph topology. By combining knowledge from high-dimensional statistics, graph theory and mathematical programming, the characterization of detection algorithms over any graph topology by their statistical properties is possible.

Aside from the statistical challenges, the computational complexity of any proposed algorithms must be addressed. Due to the combinatorial nature of graph based methods, problems can easily shift from having polynomial-time algorithms to having running times exponential in the size of the graph. The applications of graph structured inference require that any method be scalable to large graphs. As we will see, the ideal statistical procedure will be intractable, suggesting that approximation algorithms and relaxations are necessary.

## 1.1 Problem Setup

Consider a connected, possibly weighted, directed graph $G$ defined by a set of vertices $V$ ($|V| = p$) and directed edges $E$ ($|E| = m$) which are ordered pairs of vertices. Furthermore, the edges may be assigned weights, $\{W_e\}_{e \in E}$, that determine the relative strength of the interactions of the adjacent vertices. For each vertex, $i \in V$, we assume that there is an observation $y_i$ that has a Normal distribution with mean $x_i$ and variance 1. This is called the graph-structured normal means problem, and we observe one realization of the random vector

$$\mathbf{y} = \mathbf{x} + \boldsymbol{\xi}, \tag{1}$$

where $\mathbf{x} \in \mathbb{R}^p$, $\boldsymbol{\xi} \sim N(0, \mathbf{I}_{p \times p})$. The signal $\mathbf{x}$ will reflect the assumption that there is an active cluster ($C \subseteq V$) in the graph, by making $x_i > 0$ if $i \in C$ and $x_i = 0$ otherwise. Furthermore, the allowable clusters, $C$, must have a small boundary in the graph. Specifically, we assume that there are parameters $\rho, \mu$ (possibly dependent on $p$ such that the class of graph-structured activation patterns $\mathbf{x}$ is given as follows.

$$\mathcal{X} = \left\{ \mathbf{x} : \mathbf{x} = \frac{\mu}{\sqrt{|C|}} \mathbf{1}_C, C \in \mathcal{C} \right\}, \quad \mathcal{C} = \{ C \subseteq V : \text{out}(C) \leq \rho \}$$

Here $\text{out}(C) = \sum_{(u,v) \in E} W_{u,v} I\{u \in C, v \in \bar{C}\}$ is the total weight of edges leaving the cluster $C$. In other words, the set of activated vertices $C$ have a small *cut size* in the graph $G$. While we assume that the noise variance is 1 in (1), this is equivalent to the more general model in which $\mathbb{E}\xi_i^2 = \sigma^2$ with $\sigma$ known. If we wanted to consider known $\sigma^2$ then we would apply all our algorithms to $\mathbf{y}/\sigma$ and replace $\mu$ with $\mu/\sigma$ in all of our statements. For this reason, we call $\mu$ the signal-to-noise ratio (SNR), and proceed with $\sigma = 1$.

In graph-structured activation detection we are concerned with statistically testing the null against the alternative hypotheses,

$$\begin{aligned} H_0 &: \mathbf{y} \sim N(\mathbf{0}, \mathbf{I}) \\ H_1 &: \mathbf{y} \sim N(\mathbf{x}, \mathbf{I}), \mathbf{x} \in \mathcal{X} \end{aligned} \tag{2}$$

$H_0$ represents business as usual (such as sensors returning only noise) while $H_1$ encompasses all of the foreseeable anomalous activity (an elevated group of noisy sensor observations). Let a test be a mapping $T(\mathbf{y}) \in \{0, 1\}$, where 1 indicates that we reject the null. It is imperative that we control both the probability of false alarm, and the false acceptance of the null. To this end, we define our measure of risk to be

$$R(T) = \mathbb{E}_{\mathbf{0}}[T] + \sup_{\mathbf{x} \in \mathcal{X}} \mathbb{E}_{\mathbf{x}}[1 - T]$$

where $\mathbb{E}_{\mathbf{x}}$ denote the expectation with respect to $\mathbf{y} \sim N(\mathbf{x}, \mathbf{I})$. These terms are also known as the probability of type 1 and type 2 error respectively. This setting should not be confused with the Bayesian testing setup (e.g. as considered in [2, 3]) where the patterns, $\mathbf{x}$, are drawn at random. We will say that $H_0$ and $H_1$ are *asymptotically distinguished* by a test, $T$, if in the setting of large graphs, $\lim_{p \to \infty} R(T) = 0$. If such a test exists then $H_0$ and $H_1$ are *asymptotically distinguishable*, otherwise they are *asymptotically indistinguishable* (which occurs whenever the risk does not tend to 0). We will be characterizing regimes for $\mu$ in which our test asymptotically distinguishes $H_0$ from $H_1$.

Throughout the study, let the *edge-incidence matrix* of $G$ be $\nabla \in \mathbb{R}^{m \times p}$ such that for $e = (v, w) \in E$, $\nabla_{e,v} = -W_e$, $\nabla_{e,w} = W_e$ and is 0 elsewhere. For directed graphs, vertex degrees refer to $d_v = \text{out}(\{v\})$. Let $\|.\|$ denote the $\ell_2$ norm, $\|.\|_1$ be the $\ell_1$ norm, and $(\mathbf{x})_+$ be the positive components of the vector $\mathbf{x}$. Let $[p] = \{1, \ldots, p\}$, and we will be using the $o$ notation, namely if non-negative sequences satisfy $a_n/b_n \to 0$ then $a_n = o(b_n)$ and $b_n = \omega(a_n)$.

## 1.2 Contributions

Section 3 highlights what is known about the hypothesis testing problem 2, particularly we provide a regime for $\mu$ in which $H_0$ and $H_1$ are asymptotically indistinguishable. In section 4.1, we derive the graph scan statistic from the generalized likelihood ratio principle which we show to be a computationally intractable procedure. In section 4.2, we provide a relaxation of the graph scan statistic (GSS), the Lovász extended scan statistic (LESS), and we show that it can be computed with successive minimum $s - t$ cut programs (a graph cut that separates a source vertex from a sink vertex). In section 5, we give our main result, Theorem 5, that provides a type 1 error control for both test statistics, relating their performance to electrical network theory. In section 6, we show that GSS and LESS can asymptotically distinguish $H_0$ and $H_1$ in signal-to-noise ratios close to the lowest possible for some important graph models. All proofs are in the Appendix.

## 2 Related Work

**Graph structured signal processing.** There have been several approaches to signal processing over graphs. Markov random fields (MRF) provide a succinct framework in which the underlying signal is modeled as a draw from an Ising or Potts model [4, 5]. We will return to MRFs in a later section, as it will relate to our scan statistic. A similar line of research is the use of kernels over graphs. The study of kernels over graphs began with the development of diffusion kernels [6], and was extended through Green's functions on graphs [7]. While these methods are used to estimate binary signals (where $x_i \in \{0, 1\}$) over graphs, little is known about their statistical properties and their use in signal detection. To the best of our knowledge, this paper is the first connection made between anomaly detection and MRFs.

**Normal means testing.** Normal means testing in high-dimensions is a well established and fundamental problem in statistics. Much is known when $H_1$ derives from a smooth function space such as Besov spaces or Sobolev spaces[8, 9]. Only recently have combinatorial structures such as graphs been proposed as the underlying structure of $H_1$. A significant portion of the recent work in this area [10, 3, 1, 2] has focused on incorporating structural assumptions on the signal, as a way to mitigate the effect of high-dimensionality and also because many real-life problems can be represented as instances of the normal means problem with graph-structured signals (see, for an example, [11]).

**Graph scan statistics.** In spatial statistics, it is common, when searching for anomalous activity to scan over regions in the spatial domain, testing for elevated activity[12, 13]. There have been scan statistics proposed for graphs, most notably the work of [14] in which the authors scan over neighborhoods of the graphs defined by the graph distance. Other work has been done on the theory and algorithms for scan statistics over specific graph models, but are not easily generalizable to arbitrary graphs [15, 1]. More recently, it has been found that scanning over all well connected regions of a graph can be computationally intractable, and so approximations to the intractable likelihood-based procedure have been studied [16, 17]. We follow in this line of work, with a relaxation to the intractable generalized likelihood ratio test.

## 3 A Lower Bound and Known Results

In this section we highlight the previously known results about the hypothesis testing problem (2). This problem was studied in [17], in which the authors demonstrated the following lower bound, which derives from techniques developed in [3].

**Theorem 1.** *[17] Hypotheses $H_0$ and $H_1$ defined in Eq. (2) are asymptotically indistinguishable if*

$$\mu = o\left(\sqrt{\min\left\{\frac{\rho}{d_{\max}} \log\left(\frac{p d_{\max}^2}{\rho^2}\right), \sqrt{p}\right\}}\right)$$

*where $d_{\max}$ is the maximum degree of graph $G$.*

Now that a regime of asymptotic indistinguishability has been established, it is instructive to consider test statistics that do not take the graph into account (viz. the statistics are unaffected by a change in the graph structure). Certainly, if we are in a situation where a naive procedure perform near-optimally, then our study is not warranted. As it turns out, there is a gap between the performance of the natural unstructured tests and the lower bound in Theorem 1.

**Proposition 2.** *[17]* (1) *The thresholding test statistic,* $\max_{v \in [p]} |y_v|$, *asymptotically distinguishes* $H_0$ *from* $H_1$ *if* $\mu = \omega(|C| \log(p/|C|))$.
(2) *The sum test statistic,* $\sum_{v \in [p]} y_v$, *asymptotically distinguishes* $H_0$ *from* $H_1$ *if* $\mu = \omega(p/|C|)$.

As opposed to these naive tests one can scan over all clusters in $\mathcal{C}$ performing individual likelihood ratio tests. This is called the scan statistic, and it is known to be a computationally intractable combinatorial optimization. Previously, two alternatives to the scan statistic have been developed: the spectral scan statistic [16], and one based on the uniform spanning tree wavelet basis [17]. The former is indeed a relaxation of the ideal, computationally intractable, scan statistic, but in many important graph topologies, such as the lattice, provides sub-optimal statistical performance. The uniform spanning tree wavelets in effect allows one to scan over a subclass of the class, $\mathcal{C}$, but tends to provide worse performance (as we will see in section 6) than that presented in this work. The theoretical results in [17] are similar to ours, but they suffer additional log-factors.

## 4  Method

As we have noted the fundamental difficulty of the hypothesis testing problem is the composite nature of the alternative hypothesis. Because the alternative is indexed by sets, $C \in \mathcal{C}(\rho)$, with a low cut size, it is reasonable that the test statistic that we will derive results from a combinatorial optimization program. In fact, we will show we can express the generalized likelihood ratio (GLR) statistic in terms of a modular program with submodular constraints. This will turn out to be a possibly NP-hard program, as a special case of such programs is the well known knapsack problem [18]. With this in mind, we provide a convex relaxation, using the Lovász extension, to the ideal GLR statistic. This relaxation conveniently has a dual objective that can be evaluated with a binary Markov random field energy minimization, which is a well understood program. We will reserve the theoretical statistical analysis for the following section.

**Submodularity.** Before we proceed, we will introduce the reader to submodularity and the Lovász extension. (A very nice introduction to submodularity can be found in [19].) For any set, which we may as well take to be the vertex set $[p]$, we say that a function $F : \{0,1\}^p \to \mathbb{R}$ is submodular if for any $A, B \subseteq [p]$, $F(A) + F(B) \geq F(A \cap B) + F(A \cup B)$. (We will interchangeably use the bijection between $2^{[p]}$ and $\{0,1\}^p$ defined by $C \to \mathbf{1}_C$.) In this way, a submodular function experiences diminishing returns, as additions to large sets tend to be less dramatic than additions to small sets. But while this diminishing returns phenomenon is akin to concave functions, for optimization purposes submodularity acts like convexity, as it admits efficient minimization procedures. Moreover, for every submodular function there is a Lovász extension $f : [0,1]^p \to \mathbb{R}$ defined in the following way: for $\mathbf{x} \in [0,1]^p$ let $x_{j_i}$ denote the $i$th largest element of $\mathbf{x}$, then

$$f(\mathbf{x}) = x_{j_1} F(\{j_1\}) + \sum_{i=2}^{p} (F(\{j_1, \dots, j_i\}) - F(\{j_1, \dots, j_{i-1}\})) x_{j_i}$$

Submodular functions as a class is similar to convex functions in that it is closed under addition and non-negative scalar multiplication. The following facts about Lovász extensions will be important.

**Proposition 3.** *[19] Let* $F$ *be submodular and* $f$ *be its Lovász extension. Then* $f$ *is convex,* $f(\mathbf{x}) = F(\mathbf{x})$ *if* $\mathbf{x} \in \{0,1\}^p$, *and*

$$\min\{F(\mathbf{x}) : \mathbf{x} \in \{0,1\}^p\} = \min\{f(\mathbf{x}) : \mathbf{x} \in [0,1]^p\}$$

We are now sufficiently prepared to develop the test statistics that will be the focus of this paper.

### 4.1  Graph Scan Statistic

It is instructive, when faced with a class of probability distributions, indexed by subsets $\mathcal{C} \subseteq 2^{[p]}$, to think about what techniques we would use if we knew the correct set $C \in \mathcal{C}$ (which is often called oracle information). One would in this case be only testing the null hypothesis $H_0 : \mathbf{x} = \mathbf{0}$

against the simple alternative $H_1 : \mathbf{x} \propto \mathbf{1}_C$. In this situation, we would employ the likelihood ratio test because by the Neyman-Pearson lemma it is the uniformly most powerful test statistic. The maximum likelihood estimator for $\mathbf{x}$ is $\mathbf{1}_C \mathbf{1}_C^\top \mathbf{y} / |C|$ (the MLE of $\mu$ is $\mathbf{1}_C^\top \mathbf{y} / \sqrt{|C|}$) and the likelihood ratio turns out to be

$$\exp\left\{ -\frac{1}{2} \|\mathbf{y}\|^2 \right\} / \exp\left\{ -\frac{1}{2} \left\| \frac{\mathbf{1}_C \mathbf{1}_C^\top \mathbf{y}}{|C|} - \mathbf{y} \right\|^2 \right\} = \exp\left\{ \frac{(\mathbf{1}_C^\top \mathbf{y})^2}{2|C|} \right\}$$

Hence, the log-likelihood ratio is proportional to $(\mathbf{1}_C^\top \mathbf{y})^2 / |C|$ and thresholding this at $z_{1-\alpha/2}^2$ gives us a size $\alpha$ test.

This reasoning has been subject to the assumption that we had oracle knowledge of $C$. A natural statistic, when $C$ is unknown, is the generalized log-likelihood ratio (GLR) defined by $\max(\mathbf{1}_C^\top \mathbf{y})^2 / |C|$ s.t. $C \in \mathcal{C}$. We will work with the *graph scan statistic* (GSS),

$$\hat{s} = \max \frac{\mathbf{1}_C^\top \mathbf{y}}{\sqrt{|C|}} \text{ s.t. } C \in \mathcal{C}(\rho) = \{C : \text{out}(C) \le \rho\} \tag{3}$$

which is nearly equivalent to the GLR. (We can in fact evaluate $\hat{s}$ for $\mathbf{y}$ and $-\mathbf{y}$, taking a maximum and obtain the GLR, but statistically this is nearly the same.) Notice that there is no guarantee that the program above is computationally feasible. In fact, it belongs to a class of programs, specifically modular programs with submodular constraints that is known to contain NP-hard instantiations, such as the ratio cut program and the knapsack program [18]. Hence, we are compelled to form a relaxation of the above program, that will with luck provide a feasible algorithm.

### 4.2 Lovász Extended Scan Statistic

It is common, when faced with combinatorial optimization programs that are computationally infeasible, to relax the domain from the discrete $\{0,1\}^p$ to a continuous domain, such as $[0,1]^p$. Generally, the hope is that optimizing the relaxation will approximate the combinatorial program well. First we require that we can relax the constraint $\text{out}(C) \le \rho$ to the hypercube $[0,1]^p$. This will be accomplished by replacing it with its Lovász extension $\|(\nabla \mathbf{x})_+\|_1 \le \rho$. We then form the relaxed program, which we will call the *Lovász extended scan statistic* (LESS),

$$\hat{l} = \max_{t \in [p]} \max_{\mathbf{x}} \frac{\mathbf{x}^\top \mathbf{y}}{\sqrt{t}} \text{ s.t. } \mathbf{x} \in \mathcal{X}(\rho, t) = \{\mathbf{x} \in [0,1]^p : \|(\nabla \mathbf{x})_+\|_1 \le \rho, \mathbf{1}^\top \mathbf{x} \le t\} \tag{4}$$

We will find that not only can this be solved with a convex program, but the dual objective is a minimum binary Markov random field energy program. To this end, we will briefly go over binary Markov random fields, which we will find can be used to solve our relaxation.

**Binary Markov Random Fields.** Much of the previous work on graph structured statistical procedures assumes a Markov random field (MRF) model, in which there are discrete labels assigned to each vertex in $[p]$, and the observed variables $\{y_v\}_{v \in [p]}$ are conditionally independent given these labels. Furthermore, the prior distribution on the labels is drawn according to an Ising model (if the labels are binary) or a Potts model otherwise. The task is to then compute a Bayes rule from the posterior of the MRF. The majority of the previous work assumes that we are interested in the maximum a-posteriori (MAP) estimator, which is the Bayes rule for the $0/1$-loss. This can generally be written in the form,

$$\min_{\mathbf{x} \in \{0,1\}^p} \sum_{v \in [p]} -l_v(x_v | y_v) + \sum_{v \ne u \in [p]} W_{v,u} I\{x_v \ne x_u\}$$

where $l_v$ is a data dependent log-likelihood. Such programs are called graph-representable in [20], and are known to be solvable in the binary case with $s$-$t$ graph cuts. Thus, by the min-cut max-flow theorem the value of the MAP objective can be obtained by computing a maximum flow. More recently, a dual-decomposition algorithm has been developed in order to parallelize the computation of the MAP estimator for binary MRFs [21, 22].

We are now ready to state our result regarding the dual form of the LESS program, (4).

**Proposition 4.** *Let $\eta_0, \eta_1 \ge 0$, and define the dual function of the LESS,*

$$g(\eta_0, \eta_1) = \max_{\mathbf{x} \in \{0,1\}^p} \mathbf{y}^\top \mathbf{x} - \eta_0 \mathbf{1}^\top \mathbf{x} - \eta_1 \|\nabla \mathbf{x}\|_0$$

*The LESS estimator is equal to the following minimum of convex optimizations*

$$\hat{l} = \max_{t \in [p]} \frac{1}{\sqrt{t}} \min_{\eta_0, \eta_1 \geq 0} g(\eta_0, \eta_1) + \eta_0 t + \eta_1 \rho$$

$g(\eta_0, \eta_1)$ *is the objective of a MRF MAP problem, which is poly-time solvable with s-t graph cuts.*

## 5 Theoretical Analysis

So far we have developed a lower bound to the hypothesis testing problem, shown that some common detectors do not meet this guarantee, and developed the Lovász extended scan statistic from first principles. We will now provide a thorough statistical analysis of the performance of LESS. Previously, electrical network theory, specifically the effective resistances of edges in the graph, has been useful in describing the theoretical performance of a detector derived from uniform spanning tree wavelets [17]. As it turns out the performance of LESS is also dictated by the effective resistances of edges in the graph.

**Effective Resistance.** Effective resistances have been extensively studied in electrical network theory [23]. We define the combinatorial Laplacian of $G$ to be $\Delta = \mathbf{D} - \mathbf{W}$ ($\mathbf{D}_{v,v} = \text{out}(\{v\})$ is the diagonal degree matrix). A *potential difference* is any $\mathbf{z} \in \mathbb{R}^{|E|}$ such that it satisfies *Kirchoff's potential law*: the total potential difference around any cycle is 0. Algebraically, this means that $\exists \mathbf{x} \in \mathbb{R}^p$ such that $\nabla \mathbf{x} = \mathbf{z}$. The *Dirichlet principle* states that any solution to the following program gives an absolute potential $\mathbf{x}$ that satisfies Kirchoff's law:

$$\min_{\mathbf{x}} \mathbf{x}^\top \Delta \mathbf{x} \text{ s.t. } \mathbf{x}_S = \mathbf{v}_S$$

for source/sinks $S \subset [p]$ and some voltage constraints $\mathbf{v}_S \in \mathbb{R}^{|S|}$. By Lagrangian calculus, the solution to the above program is given by $\mathbf{x} = \Delta^\dagger \mathbf{v}$ where $\mathbf{v}$ is 0 over $S^C$ and $\mathbf{v}_S$ over $S$, and † indicates the Moore-Penrose pseudoinverse. The effective resistance between a source $v \in V$ and a sink $w \in V$ is the potential difference required to create a unit flow between them. Hence, the effective resistance between $v$ and $w$ is $r_{v,w} = (\delta_v - \delta_w)^\top \Delta^\dagger (\delta_v - \delta_w)$, where $\delta_v$ is the Dirac delta function. There is a close connection between effective resistances and random spanning trees. The uniform spanning tree (UST) is a random spanning tree, chosen uniformly at random from the set of all distinct spanning trees. The foundational Matrix-Tree theorem [24, 23] states that the probability of an edge, $e$, being included in the UST is equal to the edge weight times the effective resistance $W_e r_e$. The UST is an essential component of the proof of our main theorem, in that it provides a mechanism for unravelling the graph while still preserving the connectivity of the graph.

We are now in a position to state the main theorem, which will allow us to control the type 1 error (the probability of false alarm) of both the GSS and its relaxation the LESS.

**Theorem 5.** *Let* $r_{\mathcal{C}} = \max\{\sum_{(u,v) \in E : u \in C} W_{u,v} r_{(u,v)} : C \in \mathcal{C}\}$ *be the maximum effective resistance of the boundary of a cluster $C$. The following statements hold under the null hypothesis* $H_0 : \mathbf{x} = \mathbf{0}$*:*

1. *The graph scan statistic, with probability at least $1 - \alpha$, is smaller than*

$$\hat{s} \leq \left( \sqrt{r_{\mathcal{C}}} + \sqrt{\frac{1}{2} \log p} \right) \sqrt{2 \log(p-1)} + \sqrt{2 \log 2} + \sqrt{2 \log(1/\alpha)} \quad (5)$$

2. *The Lovász extended scan statistic, with probability at least $1 - \alpha$ is smaller than*

$$\hat{l} \leq \frac{\log(2p) + 1}{\sqrt{\left( \sqrt{r_{\mathcal{C}}} + \sqrt{\frac{1}{2} \log p} \right)^2 \log p}} + 2\sqrt{\left( \sqrt{r_{\mathcal{C}}} + \sqrt{\frac{1}{2} \log p} \right)^2 \log p}$$
$$+ \sqrt{2 \log p} + \sqrt{2 \log(1/\alpha)} \quad (6)$$

The implication of Theorem 5 is that the size of the test may be controlled at level $\alpha$ by selecting thresholds given by (5) and (6) for GSS and LESS respectively. Notice that the control provided for the LESS is not significantly different from that of the GSS. This is highlighted by the following Corollary, which combines Theorem 5 with a type 2 error bound to produce an information theoretic guarantee for the asymptotic performance of the GSS and LESS.

**Corollary 6.** *Both the GSS and the LESS asymptotically distinguish $H_0$ from $H_1$ if*

$$\frac{\mu}{\sigma} = \omega\left(\max\{\sqrt{r_{\mathcal{C}}\log p}, \log p\}\right)$$

To summarize we have established that the performance of the GSS and the LESS are dictated by the effective resistances of cuts in the graph. While the condition in Cor. 6 may seem mysterious, the guarantee in fact nearly matches the lower bound for many graph models as we now show.

## 6 Specific Graph Models

Theorem 5 shows that the effective resistance of the boundary plays a critical role in characterizing the distinguishability region of both the the GSS and LESS. On specific graph families, we can compute the effective resistances precisely, leading to concrete detection guarantees that we will see nearly matches the lower bound in many cases. Throughout this section, we will only be working with undirected, unweighted graphs.

Recall that Corollary 6 shows that an SNR of $\omega\left(\sqrt{r_{\mathcal{C}}\log p}\right)$ is sufficient while Theorem 1 shows that $\Omega\left(\sqrt{\rho/d_{\max}\log p}\right)$ is necessary for detection. Thus if we can show that $r_{\mathcal{C}} \approx \rho/d_{\max}$, we would establish the near-optimality of both the GSS and LESS. Foster's theorem lends evidence to the fact that the effective resistances should be much smaller than the cut size:

**Theorem 7.** *(Foster's Theorem [25, 26])*

$$\sum_{e \in E} r_e = p - 1$$

Roughly speaking, the effective resistance of an edge selected uniformly at random is $\approx (p-1)/m = d_{\text{ave}}^{-1}$ so the effective resistance of a cut is $\approx \rho/d_{\text{ave}}$. This intuition can be formalized for specific models and this improvement by the average degree bring us much closer to the lower bound.

### 6.1 Edge Transitive Graphs

An edge transitive graph, $G$, is one for which there is a graph automorphism mapping $e_0$ to $e_1$ for any pair of edges $e_0, e_1$. Examples include the $l$-dimensional torus, the cycle, and the complete graph $K_p$. The existence of these automorphisms implies that every edge has the same effective resistance, and by Foster's theorem, we know that these resistances are exactly $(p-1)/m$. Moreover, since edge transitive graphs must be $d$-regular, we know that $m = \Theta(pd)$ so that $r_e = \Theta(1/d)$. Thus as a corollary to Theorem 5 we have that both the GSS and LESS are near-optimal (optimal modulo logarithmic factors whenever $\rho/d \leq \sqrt{p}$) on edge transitive graphs:

**Corollary 8.** *Let $G$ be an edge-transitive graph with common degree $d$. Then both the GSS and LESS distinguish $H_0$ from $H_1$ provided that:*

$$\mu = \omega\left(\max\{\sqrt{\rho/d\log p}, \log p\}\right)$$

### 6.2 Random Geometric Graphs

Another popular family of graphs are those constructed from a set of points in $\mathbb{R}^D$ drawn according to some density. These graphs have inherent randomness stemming from sampling of the density, and thus earn the name *random geometric graphs*. The two most popular such graphs are *symmetric k-nearest neighbor graphs* and *$\epsilon$-graphs*. We characterize the distinguishability region for both.

In both cases, a set of points $\mathbf{z}_1, \ldots, \mathbf{z}_p$ are drawn i.i.d. from a density $f$ support over $\mathbb{R}^D$, or a subset of $\mathbb{R}^D$. Our results require mild regularity conditions on $f$, which, roughly speaking, require that $\text{supp}(f)$ is topologically equivalent to the cube and has density bounded away from zero (See [27] for a precise definition). To form a $k$-nearest neighbor graph $G_k$, we associate each vertex $i$ with a point $\mathbf{z}_i$ and we connect vertices $i, j$ if $\mathbf{z}_i$ is amongst the $k$-nearest neighbors, in $\ell_2$, of $\mathbf{z}_j$ or vice versa. In the the $\epsilon$-graph, $G_\epsilon$ we connect vertices $i, j$ if $||\mathbf{z}_i, \mathbf{z}_j|| \leq \epsilon$ for some metric $\tau$.

The relationship $r_e \approx 1/d$, which we used for edge-transitive graphs, was derived in Corollaries 8 and 9 in [27] The precise concentration arguments, which have been done before [17], lead to the following corollary regarding the performance of the GSS and LESS on random geometric graphs:

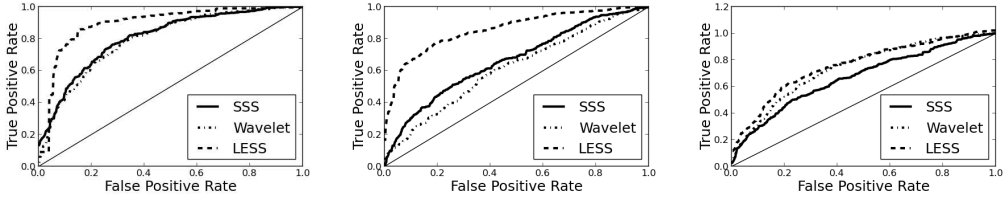

Figure 1: A comparison of detection procedures: spectral scan statistic (SSS), UST wavelet detector (Wavelet), and LESS. The graphs used are the square 2D Torus, kNN graph ($k \approx p^{1/4}$), and $\epsilon$-graph (with $\epsilon \approx p^{-1/3}$); with $\mu = 4, 4, 3$ respectively, $p = 225$, and $|C| \approx p^{1/2}$.

**Corollary 9.** *Let $G_k$ be a k-NN graph with $k/p \to 0$, $k(k/p)^{2/D} \to \infty$ and suppose the density f meets the regularity conditions in [27]. Then both the GSS and LESS distinguish $H_0$ from $H_1$ provided that:*

$$\mu = \omega \left( \max\{ \sqrt{\rho/k \log p}, \log p \} \right)$$

*If $G_\epsilon$ is an $\epsilon$-graph with $\epsilon \to 0$, $n\epsilon^{D+2} \to \infty$ then both distinguish $H_0$ from $H_1$ provided that:*

$$\mu = \omega \left( \max \left\{ \sqrt{\frac{\rho}{p\epsilon^D} \log p}, \log p \right\} \right)$$

The corollary follows immediately form Corollary 6 and the proofs in [17]. Since under the regularity conditions, the maximum degree is $\Theta(k)$ and $\Theta(p\epsilon^D)$ in $k$-NN and $\epsilon$-graphs respectively, the corollary establishes the near optimality (again provided that $\rho/d \leq \sqrt{p}$) of both test statistics.

We performed some experiments using the MRF based algorithm outlined in Prop. 4. Each experiment is made with graphs with 225 vertices, and we report the true positive rate versus the false positive rate as the threshold varies (also known as the ROC.) For each graph model, LESS provides gains over the spectral scan statistic[16] and the UST wavelet detector[17], each of the gains are significant except for the $\epsilon$-graph which is more modest.

## 7   Conclusions

To summarize, while Corollary 6 characterizes the performance of GSS and LESS in terms of effective resistances, in many specific graph models, this can be translated into near-optimal detection guarantees for these test statistics. We have demonstrated that the LESS provides guarantees similar to that of the computationally intractable generalized likelihood ratio test (GSS). Furthermore, the LESS can be solved through successive graph cuts by relating it to MAP estimation in an MRF. Future work includes using these concepts for localizing the activation, making the program robust to missing data, and extending the analysis to non-Gaussian error.

**Acknowledgments**

This research is supported in part by AFOSR under grant FA9550-10-1-0382 and NSF under grant IIS-1116458. AK is supported in part by a NSF Graduate Research Fellowship. We would like to thank Sivaraman Balakrishnan for his valuable input in the theoretical development of the paper.

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
