[Supplementary Material · nips_supp.pdf]

# 8 Appendix

Let us introduce the following notation: $W(A \to B)$ is the total weight of edges with a tail in $A$ and a head in $B \backslash A$.

**Proposition 10.** *1. out is submodular.*

*2. The Lovász extension of out is $f(\omega) = \|(\nabla \omega)_+\|$.*

*Proof. 1.* Let us partition all of the relevant edges: $w_1 = W(A \backslash B \to \overline{A \cup B}), w_2 = W(A \cap B \to \overline{A \cup B}), w_3 = W(B \backslash A \to \overline{A \cup B}), w_4 = W(A \backslash B \to B \backslash A), w_5 = W(B \backslash A \to A \backslash B), w_6 = W(A \cap B \to A \backslash B), w_7 = W(A \cap B \to B \backslash A)$. Let us then evaluate *out*,

$$\text{out}(A) + \text{out}(B) = (w_1 + w_2 + w_4 + w_7) + (w_3 + w_2 + w_5 + w_6)$$
$$\geq (w_1 + w_2 + w_3) + (w_2 + w_6 + w_7) = \text{out}(A \cup B) + \text{out}(A \cap B)$$

*2.* Let $f$ be the Lovász extension of out. Let $\mathbf{x} \in \mathbb{R}^p$, and $\{j_i\}_{i=1}^p$ be such that $x_{j_i} > x_{j_{i+1}}$. Furthermore, let $C_i = \{j_k : k > i\}$. Then, we see that $f$ takes the form,

$$f(\mathbf{x}) = \sum_{i=1}^p x_{j_i} [W(\{j_i\} \to \bar{C}_i) - W(C_i \to \{j_i\})]$$

Let us consider then the components attributable to the edge $(j_i, j_k)$; these are $W_{j_i,j_k}(x_{j_i} I(i < k) - x_{j_k} I(i < k)) = W_{j_i,j_k}(x_{j_i} - x_{j_k})_+$ because there is no contribution if $j_k \notin C_i$. This gives us our result. $\square$

*Proof of Proposition 4.* We begin with the LESS form in (4),

$$\hat{l} = \max_{t \in [p], \mathbf{x}} \frac{\mathbf{x}^\top \mathbf{y}}{\sqrt{t}} \text{ s.t. } \mathbf{x} \in \mathcal{X}(\rho, t) = \{\mathbf{x} \in [0,1]^p : \|(\nabla \mathbf{x})_+\|_1 \leq \rho, \mathbf{1}^\top \mathbf{x} \leq t\}$$

Define Lagrangian parameters $\boldsymbol{\eta} \in \mathbb{R}_+^2$ and the Lagrangian function, $L(\boldsymbol{\eta}, \mathbf{x}) = \mathbf{x}^\top \mathbf{y} - \eta_0 \mathbf{x}^\top \mathbf{1} - \eta_1 \|(\nabla \mathbf{x})_+\|_1 + \eta_0 t + \eta_1 \rho$ and notice that it is convex in $\boldsymbol{\eta}$ and concave in $\mathbf{x}$. Also, the domain $[0,1]^p$ is bounded and each domain of $L$ is non-empty closed and convex.

$$\max_{\mathbf{x} \in [0,1]^p} \inf_{\boldsymbol{\eta} \in \mathbb{R}_+^2} L(\boldsymbol{\eta}, \mathbf{x}) = \inf_{\boldsymbol{\eta} \in \mathbb{R}_+^2} \max_{\mathbf{x} \in [0,1]^p} L(\boldsymbol{\eta}, \mathbf{x})$$

This follows from a saddlepoint result in [28] (p.393 Cor. 37.3.2). All that remains is to notice that $-\mathbf{x}^\top \mathbf{y} + \eta_0 \mathbf{x}^\top \mathbf{1} + \eta_1 \|(\nabla \mathbf{x})_+\|_1$ is the Lovász extension of $-\mathbf{x}^\top \mathbf{y} + \eta_0 \mathbf{x}^\top \mathbf{1} + \eta_1 \text{out}(\mathbf{x})$ for $\mathbf{x} \in \{0,1\}^p$. Hence, by Proposition 3, there exists a minimizer that lies within $\{0,1\}^p$, and so

$$\inf_{\boldsymbol{\eta} \in \mathbb{R}_+^2} \max_{\mathbf{x} \in [0,1]^p} L(\boldsymbol{\eta}, \mathbf{x}) = \inf_{\boldsymbol{\eta} \in \mathbb{R}_+^2} g(\eta_0, \eta_1) + \eta_0 k + \eta_1 \rho$$

This follows from the fact that $\|(\nabla \mathbf{x})_+\|_1$ is equal to out$(\mathbf{x})$ for $\mathbf{x} \in \{0,1\}^p$. The program $g$ takes the form of a modular term and a cut term, which is solvable by graph cuts [29]. $\square$

## 8.1 Proof of Theorem 5

We will begin by establishing some facts about uniform spanning trees (UST). In a directed graph, a spanning tree is a tree in the graph that contains each vertex such that all the vertices but one (the root) are tails of edges in the tree. If the directed graph is not connected (i.e. there are two vertices such that there is no directed path between them) then we would have to generalize our results to a spanning forest. We will therefore assume this is not the case, for ease of presentation. Notice that in the case that we have a weighted graph, then the UST makes the probability of selecting a tree $\mathcal{T}$ proportional to the product of the constituent edge weights.

**Lemma 11.** *[30] Let $a_e \in [0,1], \forall e \in E$ and let $\mathcal{T}$ be a draw from the UST. If $Z = \sum_{e \in E} a_e I\{e \in \mathcal{T}\}$, for any $\delta \in (0,1)$,*

$$\mathbb{P}\{Z \geq (1+\delta)\mathbb{E}Z\} \leq \left(\frac{e^\delta}{(1+\delta)^{1+\delta}}\right)^{\mathbb{E}Z}$$

This implies that with probability $1-\alpha$, $Z \leq (\sqrt{\mathbb{E}Z} + \sqrt{\log(1/\alpha)})^2$ [17]. Moreover, the probability that an edge is included in $\mathcal{T}$ is its effective resistance times the edge weight, $\mathbb{P}\{e \in \mathcal{T}\} = W_e r_e$ [23].

*Proof of Theorem 5 (1).* In the following proof, for some class $\mathcal{A} \in 2^{[p]}$, let $g(\mathcal{A}) = \mathbb{E}\sup_{A \in \mathcal{A}} \frac{\mathbf{1}_A^\top \xi}{\sqrt{|A|}}$ (this is known as a Gaussian complexity). Furthermore let $\nabla_\mathcal{T}$ be the incidence matrix restricted to the edges in $\mathcal{T}$ (note that this is an unweighted directed graph). Let $\mathcal{C}(\mathcal{T}) = \{C \subset [p] : \|(\nabla_\mathcal{T}\mathbf{1}_C)_+\|_1 \leq (\sqrt{r_C} + \sqrt{\log 1/\delta})^2\}$ and $\delta > 0$ then under the UST for any $C$, $\mathbb{P}_\mathcal{T}\{C \notin \mathcal{C}(\mathcal{T})\} \leq \delta$. (This follows from Lemma 11.)

$$\mathbb{E}_\xi \sup_{C \in \mathcal{C}} \frac{\xi^\top \mathbf{1}_C}{\sqrt{|C|}} = \mathbb{E}_\xi \sup_{C \in \mathcal{C}} \mathbb{E}_\mathcal{T} \frac{\xi^\top \mathbf{1}_C}{\sqrt{|C|}} [\mathbf{1}\{C \in \mathcal{C}(\mathcal{T})\} + \mathbf{1}\{C \notin \mathcal{C}(\mathcal{T})\}]$$

$$\leq \mathbb{E}_\xi \sup_{C \in \mathcal{C}} \left[ \mathbb{E}_\mathcal{T} \mathbf{1}\{C \in \mathcal{C}(\mathcal{T})\} \sup_{C' \in \mathcal{C}(\mathcal{T})} \frac{\xi^\top \mathbf{1}_{C'}}{\sqrt{|C'|}} + \mathbb{E}_\mathcal{T} \mathbf{1}\{C \notin \mathcal{C}(\mathcal{T})\} \sup_{C' \in 2^{[p]}} \frac{\xi^\top \mathbf{1}'_C}{\sqrt{|C'|}} \right]$$

$$\leq \mathbb{E}_\xi \sup_{C \in \mathcal{C}} \left[ \mathbb{E}_\mathcal{T} \sup_{C' \in \mathcal{C}(\mathcal{T})} \frac{\xi^\top \mathbf{1}_{C'}}{\sqrt{|C'|}} + \mathbb{E}_\mathcal{T} \mathbf{1}\{C \notin \mathcal{C}(\mathcal{T})\} \sup_{C' \in 2^{[p]}} \frac{\xi^\top \mathbf{1}'_C}{\sqrt{|C'|}} \right]$$

$$\leq \mathbb{E}_\xi \left[ \mathbb{E}_\mathcal{T} \sup_{C' \in \mathcal{C}(\mathcal{T})} \frac{\xi^\top \mathbf{1}_{C'}}{\sqrt{|C'|}} + \sup_{C \in \mathcal{C}} \mathbb{P}_\mathcal{T}\{C \notin \mathcal{C}(\mathcal{T})\} \sup_{C' \in 2^{[p]}} \frac{\xi^\top \mathbf{1}'_C}{\sqrt{|C'|}} \right]$$

$$\leq \mathbb{E}_\mathcal{T} g(\mathcal{C}(\mathcal{T})) + g(2^{[p]}) \sup_{C \in \mathcal{C}} \mathbb{P}_\mathcal{T}\{C \notin \mathcal{C}(\mathcal{T})\}$$

For any $\mathcal{T}$, $|\mathcal{C}(\mathcal{T})| \leq (p-1)^{(\sqrt{r_C}+\sqrt{\log 1/\delta})^2}$ because $\mathcal{T}$ is unweighted. By Gaussianity and the fact that $\mathbb{E}(\mathbf{1}_C^\top \xi/\sqrt{|C|})^2 = 1$,

$$g(\mathcal{C}(\mathcal{T})) \leq \sqrt{2 \log |\mathcal{C}(\mathcal{T})|} \leq \sqrt{2(\sqrt{r_C} + \sqrt{\log 1/\delta})^2 \log(p-1)}$$

Furthermore, $g(2^{[p]}) \leq a\sqrt{p}$ where $a = \sqrt{2\log 2}$. Setting $\delta = p^{-1/2}$ we have the following bound on the Gaussian complexity,

$$g(\mathcal{C}) \leq (\sqrt{r_C} + \sqrt{\frac{1}{2}\log p})\sqrt{2\log(p-1)} + a$$

By Cirelson's theorem [31], with probability at least $1 - \alpha$,

$$\sup_{C \in \mathcal{C}} \frac{\xi^\top \mathbf{1}_C}{\sqrt{|C|}} \leq g(\mathcal{C}) + \sqrt{2\log(1/\alpha)}$$

$\square$

*Proof of Theorem 5 (2).* Let $\mathcal{X}(\mathcal{T}) = \{\mathbf{x} \in [0,1]^p : \|(\nabla_\mathcal{T}\mathbf{x})_+\|_1 \leq (\sqrt{r_\mathcal{X}} + \sqrt{\log 1/\delta})^2\}$. It remains the case that, by the previous Lemma 11, $\mathbb{P}\{\|(\nabla_\mathcal{T}\mathbf{x})_+\|_1 \geq (\sqrt{r_\mathcal{X}} + \sqrt{\log 1/\delta})^2\} \leq \delta$, where $r_\mathcal{X} = \{\max \sum_{(j,i) \in E} W_e r_e (x_i - x_j)_+ : \mathbf{x} \in \mathcal{X}\}$.

$$\mathbb{E}_\xi \hat{l} = \mathbb{E}_\xi \sup_{t \in [p], \mathbf{x} \in \mathcal{X}(\rho,t)} \frac{\xi^\top \mathbf{x}}{\sqrt{t}} = \mathbb{E}_\xi \sup_{t \in [p], \mathbf{x} \in \mathcal{X}(\rho,t)} \mathbb{E}_\mathcal{T} \frac{\xi^\top \mathbf{x}}{\sqrt{t}} [\mathbf{1}\{\mathbf{x} \in \mathcal{X}(\mathcal{T})\} + \mathbf{1}\{\mathbf{x} \notin \mathcal{X}(\mathcal{T})\}]$$

$$\leq \mathbb{E}_\xi \sup_{t \in [p], \mathbf{x} \in \mathcal{X}(\rho,t)} \left[ \mathbb{E}_\mathcal{T} \mathbf{1}\{\mathbf{x} \in \mathcal{X}(\mathcal{T})\} \sup_{\mathbf{x}' \in \mathcal{X}(\mathcal{T}), \mathbf{1}^\top \mathbf{x}' \leq t} \frac{\xi^\top \mathbf{x}'}{\sqrt{t}} + \mathbb{E}_\mathcal{T} \mathbf{1}\{\mathbf{x} \notin \mathcal{X}(\mathcal{T})\} \sup_{\mathbf{x}' \in [0,1]^p, \mathbf{1}^\top \mathbf{x}' \leq t} \frac{\xi^\top \mathbf{x}'}{\sqrt{t}} \right]$$

$$\leq \mathbb{E}_\xi \sup_{t \in [p], \mathbf{x} \in \mathcal{X}(\rho,t)} \left[ \mathbb{E}_\mathcal{T} \sup_{\mathbf{x}' \in \mathcal{X}(\mathcal{T}), \mathbf{1}^\top \mathbf{x}' \leq t} \frac{\xi^\top \mathbf{x}'}{\sqrt{t}} + \mathbb{E}_\mathcal{T} \mathbf{1}\{\mathbf{x} \notin \mathcal{X}(\mathcal{T})\} \sup_{\mathbf{x}' \in [0,1]^p, \mathbf{1}^\top \mathbf{x}' \leq t} \frac{\xi^\top \mathbf{x}'}{\sqrt{t}} \right]$$

$$\leq \mathbb{E}_\mathcal{T} \mathbb{E}_\xi \sup_{t \in [p], \mathbf{x} \in \mathcal{X}(\mathcal{T}), \mathbf{1}^\top \mathbf{x} \leq t} \frac{\xi^\top \mathbf{x}}{\sqrt{t}} + \sup_{\mathbf{x} \in \mathcal{X}(\rho)} \mathbb{P}_\mathcal{T}\{\mathbf{x} \notin \mathcal{X}(\mathcal{T})\} \mathbb{E}_\xi \sup_{t \in [p], \mathbf{x} \in [0,1]^p, \mathbf{1}^\top \mathbf{x} \leq t} \frac{\xi^\top \mathbf{x}}{\sqrt{t}}$$

These follow from Jensen's inequality and Fubini's theorem.

**Claim 12.**

$$\mathbb{E}_\xi \sup_{t\in[p],\mathbf{x}\in[0,1]^p,\mathbf{1}^\top\mathbf{x}\leq t} \frac{\xi^\top\mathbf{x}}{\sqrt{t}} \leq \sqrt{2p\log 2}$$

We will proceed to prove the above claim. In words it follows from the fact that solutions to the program are integral by the generic chaining.

$$\mathbb{E}_\xi \sup_{t\in[p],\mathbf{x}\in[0,1]^p,\mathbf{1}^\top\mathbf{x}\leq t} \frac{\xi^\top\mathbf{x}}{\sqrt{t}} = \mathbb{E}_\xi \sup_{t\in[p]} \frac{1}{\sqrt{t}} \sup_{\mathbf{x}\in[0,1]^p:\mathbf{1}^\top\mathbf{x}\leq t} \xi^\top\mathbf{x}$$

$$= \mathbb{E}_\xi \sup_{t\in[p]} \frac{1}{\sqrt{t}} \sup_{\mathbf{x}\in\{0,1\}^p:\mathbf{1}^\top\mathbf{x}\leq t} \xi^\top\mathbf{x} = \mathbb{E}_\xi \sup_{\mathbf{x}\in\{0,1\}^p} \frac{\xi^\top\mathbf{x}}{\|\mathbf{x}\|} \leq \sqrt{2p\log 2}$$

The second equality holds because the solution to the optimization with $t$ fixed is the top $t$ coordinates of $\xi$. The third equality holds because $\mathbf{x}\in\{0,1\}^p$ and so $\mathbf{1}^\top\mathbf{x}$ is integer. Hence, if $\mathbf{x}$ is a solution for the objective with $t$ fixed and $\mathbf{1}^\top\mathbf{x} < t$ then it holds for the objective with $t-1$, and the overall objective is increased. Thus at the optimum, $\|\mathbf{x}\| = \sqrt{\mathbf{1}^\top\mathbf{x}} = \sqrt{t}$.

**Claim 13.** *Denote $r = (\sqrt{r_\mathcal{X}} + \sqrt{\frac{1}{2}\log p})^2$. For any spanning tree $\mathcal{T}$,*

$$\mathbb{E}_\xi \sup_{t\in[p],\mathbf{x}\in\mathcal{X}(\mathcal{T}),\mathbf{1}^\top\mathbf{x}\leq t} \frac{\xi^\top\mathbf{x}}{\sqrt{t}} \leq \frac{\log(2p)+1}{\sqrt{r\log p}} + 2\sqrt{r\log p}$$

This will follow from weak duality and a clever choice of dual parameters.

$$\sup_{t\in[p]} \frac{1}{\sqrt{t}} \sup_{\mathbf{x}\in\mathcal{X}(\mathcal{T}),\mathbf{1}^\top\mathbf{x}\leq t} \xi^\top\mathbf{x}$$

$$= \sup_{t\in[p]} \frac{1}{\sqrt{t}} \sup_{\mathbf{x}\in[0,1]^p} \inf_{\eta\geq 0} \xi^\top\mathbf{x} - \eta_0\mathbf{1}^\top\mathbf{x} - \eta_1\|(\nabla_\mathcal{T}\mathbf{x})_+\|_1 + \eta_0 t + \eta_1 r$$

$$\leq \sup_{t\in[p]} \frac{1}{\sqrt{t}} \sup_{\mathbf{x}\in\{0,1\}^p} \xi^\top\mathbf{x} - \mathbf{1}^\top\mathbf{x}\sqrt{\frac{r}{t}\log p} - \|(\nabla_\mathcal{T}\mathbf{x})_+\|_1\sqrt{\frac{t}{r}\log p} + 2\sqrt{rt\log p}$$

The above display follows by selecting $\eta_0 = \sqrt{\frac{r}{t}\log p}$ and $\eta_1 = \sqrt{\frac{t}{r}\log p}$ and using Prop. 3.

$$= \sup_{k\in[p]} \sup_{\mathbf{x}\in\{0,1\}^p:\mathrm{out}(\mathbf{x})=k} \sup_{t\in[p]} \frac{\xi^\top\mathbf{x}}{\sqrt{t}} - \frac{\mathbf{1}^\top\mathbf{x}}{t}\sqrt{r\log p} - k\sqrt{\frac{1}{r}\log p} + 2\sqrt{r\log p}$$

$$\leq \sup_{k\in[p]} \sup_{\mathbf{x}\in\{0,1\}^p:\mathrm{out}(\mathbf{x})=k} \frac{(\xi^\top\mathbf{x})^2}{4\|\mathbf{x}\|^2\sqrt{r\log p}} - k\sqrt{\frac{1}{r}\log p} + 2\sqrt{r\log p}$$

The above display follows from the fact that for any $a, b > 0$, $\sup_{t\in\mathbb{R}} at - bt^2 = a^2/(4b)$. We know that with probability at least $1-\alpha$ for all $k\in[p]$,

$$\sup_{\mathbf{x}\in\{0,1\}^p,\mathrm{out}(\mathbf{x})=k} \left|\frac{\xi^\top\mathbf{x}}{\|\mathbf{x}\|}\right| \leq \sqrt{2k\log p} + \sqrt{2\log(2p/\alpha)}$$

So we can bound the above,

$$\sup_{t\in[p]} \frac{1}{\sqrt{t}} \sup_{\mathbf{x}\in\mathcal{X}(\mathcal{T}),\mathbf{1}^\top\mathbf{x}\leq t} \xi^\top\mathbf{x} \leq \sup_{k\in[p]} \frac{(\sqrt{2k\log p} + \sqrt{2\log(2p/\alpha)})^2}{4\sqrt{r\log p}} - k\sqrt{\frac{1}{r}\log p} + 2\sqrt{r\log p}$$

$$= \sup_{k\in[p]} \frac{\sqrt{k\log(2p/\alpha)}}{\sqrt{r}} - \frac{k}{2}\sqrt{\frac{\log p}{r}} + \frac{\log(2p/\alpha)}{2\sqrt{r\log p}} + 2\sqrt{r\log p}$$

$$\leq \frac{\log(2p/\alpha)}{2\sqrt{r\log p}} + \frac{\log(2p/\alpha)}{2\sqrt{r\log p}} + 2\sqrt{r\log p}$$

$$= \frac{\log(2p/\alpha)}{\sqrt{r\log p}} + 2\sqrt{r\log p}$$

Any random variable $Z$ that satisfies $Z \leq a + b \log(1/\alpha)$ with probability $1 - \alpha$ for any $\alpha > 0$ for $a, b \geq 0$ also satisfies $\mathbb{E}Z \leq a + b$. Hence,

$$\mathbb{E}_\xi \sup_{t \in [p]} \frac{1}{\sqrt{t}} \sup_{\mathbf{x} \in \mathcal{X}(\mathcal{T}), \mathbf{1}^\top \mathbf{x} \leq t} \xi^\top \mathbf{x} \leq \frac{\log(2p) + 1}{\sqrt{r \log p}} + 2\sqrt{r \log p}$$

Combining all of these results and using Cirelson's theorem [31],

$$\hat{l} \leq \frac{\log(2p) + 1}{\sqrt{\left(\sqrt{r_\mathcal{X}} + \sqrt{\frac{1}{2} \log p}\right)^2 \log p}} + 2\sqrt{\left(\sqrt{r_\mathcal{X}} + \sqrt{\frac{1}{2} \log p}\right)^2 \log p}$$
$$+ \sqrt{2 \log 2} + \sqrt{2 \log(1/\alpha)}$$

All that remains to be show is that $r_\mathcal{X} = r_\mathcal{C}$. This can be seen by constructing the level sets of $\mathbf{x} \in [0,1]^p$ and noticing that $\sum_{(i,j) \in E} W_e r_e (x_j - x_i)_+$ is piecewise linear in the levels. Thus, we can draw a contradiction from the supposition that the levels are not in $\{0, 1\}$. $\qquad \square$

*Proof of Corollary 6.* We will argue that with high probability, under $H_1$ the GSS and LESS are large. For the analysis of both the GSS and the LESS, let

$$\mathbf{x}^* = \mathbf{1}_C, \quad t^* = |C|$$

Then both the GSS and LESS are lower bounded by

$$\frac{\mathbf{1}_C^\top \mathbf{y}}{\sqrt{|C|}} = \mu + \frac{\mathbf{1}_C^\top \xi}{\sqrt{|C|}} \sim \mathcal{N}(\mu, 1)$$

Hence, under $H_1$, with probability $1 - \alpha$, the GSS and LESS are larger than $\mu - \sqrt{2 \log(1/\alpha)}$. The Corollary follows by comparing this to the guarantee in Theorem 5. $\qquad \square$