[Reviews · NeurIPS 2013]

Submitted by Assigned_Reviewer_3

This paper presents a new algorithm to detect activation in a weighted graph, as well
as some theoretical analysis regarding minimal detection levels of this algorithm
as well as others.

The problem considered - detecting activation in graphs is (relatively)
new and interesting. Furthermore, the new solution proposed by the authors
is to the best of my knowledge highly original, and may be of applicability in other
problems.

That said, the paper is not very clearly written, and is not easy to read and follow.

Detailed comments are as follows:
p.1 L36 - Detection is broader than as written in first sentence of intro. Consider
for example change detection, outlier detection etc. In many such problems the null is
not merely noise.

p.2 L57-72: Not clear why authors spend these lines presenting these applications.
On first read I thought they actually had data from these applications and would
show the results of their algorithms. The wording on line 55 is quite misleading
"we will examine two real-world examples". In fact, can the authors point to links
where such data is available in the public domain ?

p.2 L85: is C required to be a single connected component ? Does not seem so from definition,
and constraint that out(C) \leq rho means that for sufficiently large rho C can be a
collection of a quite large number of singletons.

p2 L87 number of edges --> total weight ?

p.2 L100 typically one cannot control both types of error, but rather tries to maximize
probability of detection while fixing the false alarm rate.

p2 L106 "instead of the test being randomized" not clear.

p3 top - it is not clear that as p\to\infty there is a 0/1 limiting law of perfect
distinguishability or indistinguishability (type I plus type II errors tending to 0 or to 1).

p3 L135 "used to localize binary signals over graphs" not clear.

p4 L165 "that do not take the graph into account" is not clear.

p4 L172-177 - whole paragraph is not very clear. Also, w.r.t. proposition 2 and the problem formulation,
I would like to point out that there may be a difference between detection
of presence of activation in the graph, and estimation of which nodes where activated.
See for example work by Y. Ingster (1997) and by Donoho and Jin, Annals of Statistics, 2004.
I presume there may be a similar phenomena here, unless there is a constraint that C
is connected ?

p.4 L177 tends to provide worse performance - is this an empirical observation or is there
a mathematically precise statement.

p 5 can the authors comment on the complexity of the LESS algorithm as a function of p
rho etc.

p.7 how does r_C depend on rho ?


Summary: Very interesting paper regarding detection of activation in a weighted graph.

Submitted by Assigned_Reviewer_5

The authors propose a novel scan statistic LESS that identifies anomalous activities in the graph. The paper is fairly theoretical and provides several important results and an especially important Type 1 error analysis. The only problem is that it tries to pack too much and some of the details and the connections are lost, such as the importance of the Lovasz relaxation is not really clear. Still the paper is fairly well written even if very dense and provides a set of novel results. Impressively, the proposed LESS statistic performs well experimentally, though the results are not really discussed in detail, for example why would it not perform well on an \epsilon-graph and what are the conditions when it would perform well? I find it to be an important and interesting contribution though it's somewhat outside of my area of expertise and I hope that the authors will publish an extended version in a journal.
Summary: Very technical paper, several interesting and important results, perhaps too dense for 9 pages, would be nice to see an extended version in a journal.

Submitted by Assigned_Reviewer_6

The problem of anomaly detection in graphs is considered. The basic
hypothesis test is to distinguish between two hypotheses, N(0,I) and
N(x,I) on the graph. The paper looks at related work, especially
graph scan statistics. Lower bounds and simpler test are considered.
Next, the authors propose a method based on an oracle scan statistic.
By using a relaxation of the scan statistic (LESS), the authors find a
solvable problem. Proposition 4 gives the dual formulation and
solution method. Theoretical analysis of LESS gives asymptotic
performance of GSS and LESS. Finally, the method is applied to a
variety of synthetic graphs. Results show LESS outperforms other
methods in many cases in terms of ROC performance.

Quality
The paper is complete and the derived results and statistics appear
reasonable (although this reviewer did not check them in detail).
Experimental results in Figure 1 support the claims of the paper and
show significant improvement over other methods.

Clarity
The methods and techniques are clearly described. One point
of improvement--the paper has an informal style in some areas which is
distracting. Some phrases could be rewritten; e.g., line 068, "I
interact with my officemates quite often ..."

Originality
This work appears to be distinct from prior efforts.

Significance
The detection of anomalies in graphs is an area of ongoing interest.
The contribution of the authors is a significant advance over the
prior referenced work. One drawback is that the prior referenced work
in [16], [17] is still in preprint form. Is there prior published
work that could be added?
Summary: The paper describes completely the detection of anomalies on graphs using a scan statistic. Experimental results demonstrate the effectiveness of the methods.
Author Feedback

Author rebuttal: First and foremost, we thank the reviewers for their careful analysis of our paper.

Reviewer 1: The points of confusion raised by Reviewer 1 will be addressed in the camera ready version, specifically we will make it clear that the examples presented are only for the purpose of motivating the work.
Regarding the comment about 0/1 law, it is indeed true that there is a regime in which the risk converges to something not 0 or 1, but in this work we consider this case to also be indistinguishable. C is not required to be a connected component and indeed if rho is large relative to the cluster size, then the cluster can consist of singletons. However, for clusters with small cut size to cluster size ratio, the method takes advantage of the cluster structure.
Reviewer 1 is correct that there may be different SNR regimes for detection and localization of the cluster, C, we are not trying to make any statements about localization in this paper. The comment about performance of spanning tree wavelet detector proposed in [17] is a quantitative statement. In comparison to Theorem 5 of this paper, the bounds in [17] contain an additional log factor.
We will elaborate on the complexity of the LESS, which is well known because it can be solved with max-flow algorithms. Finally, as discussed in section 6, for many graphs $r_C \approx rho/d_max$.

Reviewer 2: We believe the reason Lovasz extension is important is that it yields an $\ell_1$ relaxation of the combinatorial scan statistic (which involves an $\ell_0$ constraint on the number of edges that differ in value). The spectral scan statistic proposed in [16] replaces the binary constraint on edge differences to an $\ell_2$ constraint. Since $\ell_1$ is a tighter relaxation of an $\ell_0$ constraint than $\ell_2$, Lovasz extension can be expected to outperform the spectral scan statistic [16]. We will add a discussion to this effect.

The reviewer also mentioned the reduced performance of the LESS on the epsilon-random graph. The SNR in this example is smaller than that of the other graphs (3 as opposed to 4), so really the question is why does the wavelet estimator work as well as the LESS in this case. We suspect this might be the case since wavelet detector is an $\ell_0$ detector in a transform domain. However, we don't have a concrete answer to this at the moment, but are exploring why this might be the case.

Reviewer 3: [16] and [17] were published in proceedings of AISTATS 2013 recently. We will update the references.